# Integrating structure-based machine learning and co-evolution to investigate specificity in plant sesquiterpene synthases

Janani Durairaj[1]*, Elena Melillo[2], Harro J. Bouwmeester[3], Jules Beekwilder[4,5], Dick de Ridder[1], Aalt D. J. van Dijk[1,6]*

1 Bioinformatics Group, Department of Plant Sciences, Wageningen University and Research, Wageningen, The Netherlands, 2 Isobionics, Geleen, The Netherlands, 3 Swammerdam Institute for Life Sciences, University of Amsterdam, Amsterdam, The Netherlands, 4 Bioscience, Wageningen Plant Research, Wageningen University and Research, Wageningen, The Netherlands, 5 Laboratory of Plant Physiology, Department of Plant Sciences, Wageningen University and Research, Wageningen, The Netherlands, 6 Biometris, Department of Plant Sciences, Wageningen University and Research, Wageningen, The Netherlands

* janani.durairaj@wur.nl (JD); aaltjan.vandijk@wur.nl (ADJD)

**Data Availability Statement:** The authors confirm that all data underlying the findings are fully available without restriction. All relevant data and

## Abstract

Sesquiterpene synthases (STSs) catalyze the formation of a large class of plant volatiles called sesquiterpenes. While thousands of putative STS sequences from diverse plant species are available, only a small number of them have been functionally characterized. Sequence identity-based screening for desired enzymes, often used in biotechnological applications, is difficult to apply here as STS sequence similarity is strongly affected by species. This calls for more sophisticated computational methods for functionality prediction. We investigate the specificity of precursor cation formation in these elusive enzymes. By inspecting multi-product STSs, we demonstrate that STSs have a strong selectivity towards one precursor cation. We use a machine learning approach combining sequence and structure information to accurately predict precursor cation specificity for STSs across all plant species. We combine this with a co-evolutionary analysis on the wealth of uncharacterized putative STS sequences, to pinpoint residues and distant functional contacts influencing cation formation and reaction pathway selection. These structural factors can be used to predict and engineer enzymes with specific functions, as we demonstrate by predicting and characterizing two novel STSs from *Citrus bergamia*.

## Author summary

Predicting enzyme function is a popular problem in the bioinformatics field that grows more pressing with the increase in protein sequences, and more attainable with the increase in experimentally characterized enzymes. Terpenes and terpenoids form the largest classes of natural products and find use in many drugs, flavouring agents, and perfumes. Terpene synthases catalyze the biosynthesis of terpenes via multiple cyclizations and carbocation rearrangements, generating a vast array of product skeletons. In this

code are available at https://git.wageningenur.nl/durai001/sts_cation_prediction and in the Supporting information files. The three novel enzyme sequences discussed in the text are available from the NCBI GenBank database with accessions numbers MT636927, MT636928 and MW384854.

**Funding:** This work is supported by the research programme Novel Enzymes for Flavour and Fragrance with grant number TTW 15043 (to HJB), which is financed by the Netherlands Organisation for Scientific Research (NWO, https://www.nwo.nl). The funders had no role in study design, data collection and analysis, decision to publish, or preparation of the manuscript.

**Competing interests:** I have read the journal's policy and the authors of this manuscript have the following competing interests: JB and EM are currently employed by Isobionics, a company involved in using terpenes for fragrance and flavour products.

work, we present a three-pronged computational approach to predict carbocation specificity in sesquiterpene synthases, a subset of terpene synthases with one of the highest diversities of products. Using homology modelling, machine learning and co-evolutionary analysis, our approach combines sparse structural data, large amounts of uncharacterized sequence data, and the current set of experimentally characterized enzymes to provide insight into residues and structural regions that likely play a role in determining product specificity. Similar techniques can be re-purposed for function prediction and enzyme engineering in many other classes of enzymes.

## Introduction

One of the largest and most structurally diverse family of plant-derived natural products is the isoprenoid or terpenoid family, with over 60,000 members comprising mono-, sesqui-, di-, tri-, and sesterterpenes, along with steroids and carotenoids [1]. These phytochemicals serve plants in defence against pathogens or herbivores and as attractants of pollinators [2]. They are also of high economic value to humankind due to their widespread use in pharmaceutical agents, insecticides, preservatives, fragrances, and flavoring [3]. The immense diversity of the terpenoid family derives from the polymerization and rearrangement of a varying number of simple 5-carbon isoprenoid units. Monoterpenes are 10-carbon (C10) compounds built up of two such units, sesquiterpenes are composed of three and hence are C15 compounds, diterpenes (C20) are composed of four, and so on. Sesquiterpenes are especially interesting due to their high diversity. Their formation is catalyzed from the C15 substrate, farnesyl pyrophosphate (FPP), by sesquiterpene synthases (STSs), a class of enzymes found in plants, fungi and bacteria [4].

Recently, we published a database of over 250 experimentally characterized STSs from over one hundred plant species, collectively responsible for the formation of over a hundred different sesquiterpenes [5]. These compounds all derive from the same substrate, FPP, through a branching tree of reactions such as cyclizations, hydride shifts, methyl shifts, rearrangements, re- and de-protonations to give rise to the immense existing variety in sesquiterpene structures. Apart from the functionally characterized STSs in the database, there are thousands of putative STSs in sequenced plant genomes and transcriptomes whose product specificity is unknown. In addition, many STSs in our database are multi-product enzymes, further complicating the matter of product specificity prediction. As a first contribution, we show that multi-product STSs usually catalyze products specific to a single pathway, indicating selectivity towards one precursor cation. Finding residue positions related to this cation choice across all STSs can reveal important aspects of the underlying mechanisms. However, our previous sequence-based analysis showed that these enzymes are very diverse and sequence similarity is heavily influenced by phylogeny [5]. While an approach using hidden Markov models derived from sequences is available to predict what kind of terpene synthase (mono-, di-, tri-, sesqui- etc.) a particular enzyme may be [6], this kind of sequence-based grouping was not seen within STSs making products derived from a particular cation or cyclization [5]. As a result, previous studies directed at identifying determinants of catalytic specificity in STSs mainly used mutational approaches between and within a few enzymes from the same or closely related species [7–10]. While such approaches have been successful in finding residues influencing product specificity, their small scale in light of the large diversity of STSs makes it likely that they miss aspects shared across all plant STSs. However, terpene synthases across plants, animals, fungi, and bacteria all share a common structural fold [11]. Protein structures typically evolve at a

slower pace than sequences, which means they can contain a wealth of information not easily retrieved from the corresponding sequences.

Here, we combine homology modelling to incorporate STS structural information and machine learning to tease out contributions of different residues to cation specificity. We show that structure-based prediction performs well across all plant species, including on STS enzymes that were published recently and were not used for the construction of the predictor. Such structure- or model-based machine learning has been explored before in other enzyme families and prediction tasks [12–15], and is challenging. One major challenge is the immense number of features produced, as each protein has many hundreds of residues, each of which has its own set of structural features. This poses a problem in cases like the current one, where labeled, experimentally characterized data is sparse. Here we used a novel hierarchical classification approach where many classifiers are first trained on each feature across all residues, after which the most predictive residues are selected. The final classifier is only trained on the feature values of these predictive residues. Thus, we are able to prune noisy and irrelevant features in order to pinpoint residue positions correlating with cation specificity. These selected residues are likely intrinsically linked to the catalytic mechanism of an STS and contribute to the enzymatic formation of the precursor cation. Many of these residues are also not found when relying on sequence-derived features alone, emphasizing the importance of structure in understanding catalytic activity.

In addition, while the current characterized sequence space may be small, there are many thousands of uncharacterized putative terpene synthases whose sequences can provide valuable information about evolution and conservation, especially in regions where reliable structural information is not available. A correlated mutations analysis on all putative terpene synthases indicates co-evolving residue partners for our set of cation-specific residues which are implicated in shared functional activity (such as intermediate binding or coordination), favoring their co-evolution. Examining these residues and pairs in the context of each other and co-crystallized substrate analogs reveals important aspects of the STS reaction mechanism.

Apart from the independent test set of recently characterized enzymes, we also present a use-case of our predictor for STS specificity screening by predicting and characterizing bisabolyl cation synthases from *Citrus bergamia*, which further demonstrated the accuracy of the predictor. As the number of experimentally characterized STSs grows, this accuracy will further increase, potentially allowing for more fine-grained product specificity prediction.

The three-pronged approach presented here combines a modest amount of labelled sequence data, a very small amount of experimental structure data, and large amounts of unlabeled sequence data using homology modelling, interpretable machine learning, and co-evolutionary analysis to predict and investigate the underlying mechanisms of cation specificity in STSs. This approach can also be useful for exploring specificity in other enzyme families with characteristics similar to the STSs.

## Results and discussion

### Sesquiterpene synthases follow a single branch of the reaction tree

The reaction cascade of an STS can take two directions. As is depicted in Fig 1, all reactions are initiated by a metal-mediated removal of the diphosphate anion in the (*E,E*)-FPP substrate, leading to the formation of a transoid (2*E*,6*E*)-farnesyl cation (farnesyl cation). The farnesyl cation may then isomerize to form a cisoid (2*Z*,6*E*)-farnesyl cation (nerolidyl cation). These two cations may be quenched by water or undergo a proton loss to form acyclic products (acyclic-F and acyclic-N). However, both farnesyl and nerolidyl cations can undergo cyclization at the C10-C11 bond, while the nerolidyl cation can also cyclize at the C6-C7 bond. The resulting

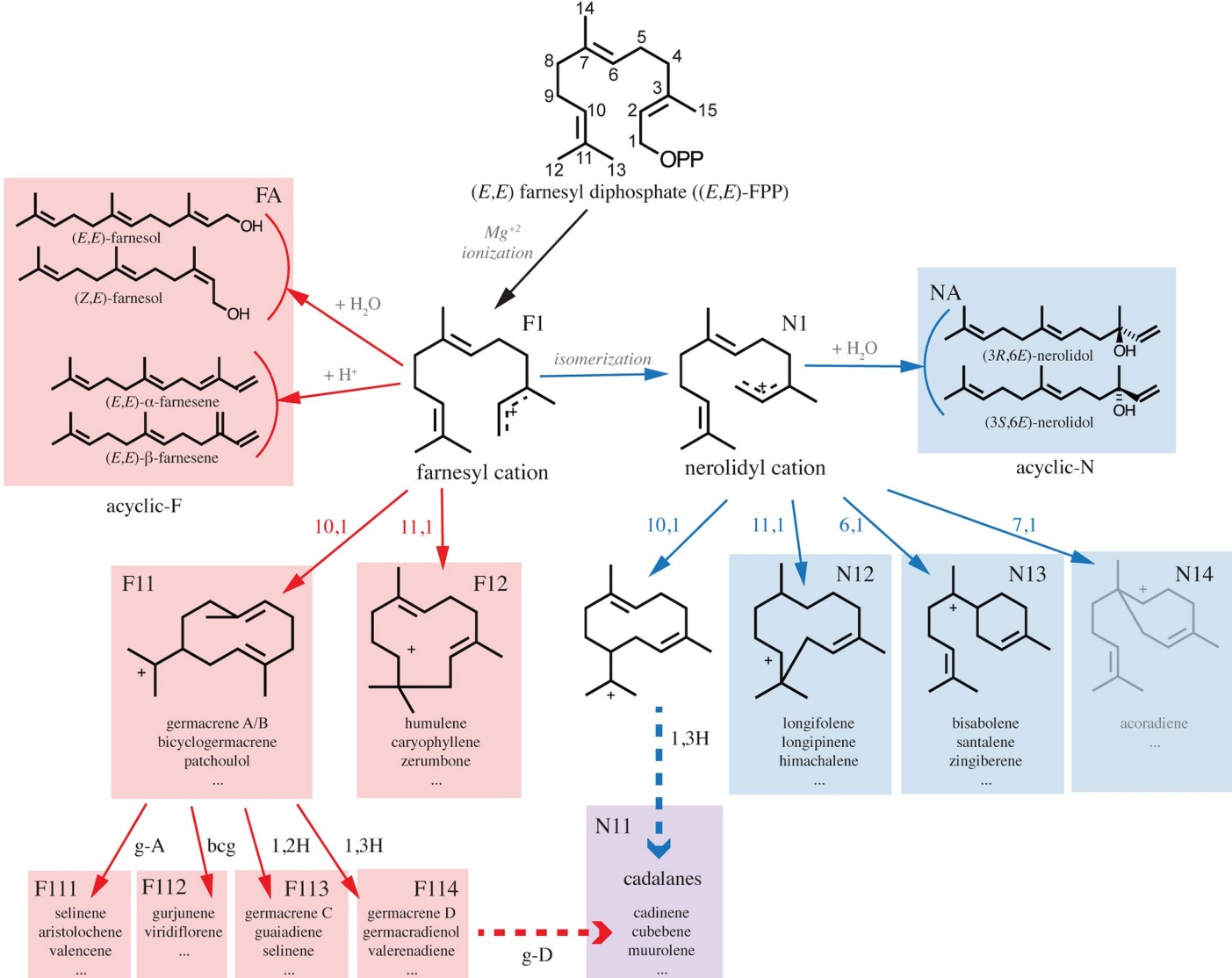

**Fig 1. The reaction mechanism of sesquiterpene production starts with farnesyl diphosphate ((E,E)-FPP).** Loss of the diphosphate moiety (OPP) leads to farnesyl cation formation. The farnesyl cation can subsequently be converted to the nerolidyl cation. Acyclic sesquiterpenes (acyclic-F and acyclic-N) are formed from these two cations by proton loss or reaction with water molecules. Possible cyclizations for both cations are indicated in the figure. The subsequently formed cyclic cations undergo modifications and rearrangements to form cyclic sesquiterpenes. Some of these sesquiterpenes (g-A and bcg) themselves act as neutral intermediates which can be re-protonated and undergo further reactions to form more products. Products are also formed from specific charged intermediates such as a 1,2- or 1,3-hydride shift of the 10,1-cyclized farnesyl cation (1,2H, 1,3H) and the cadalane skeleton (cadalanes), which can be formed via either of the two precursor cations, or via acid-induced rearrangement of germacrene D. The 7,1-cyclization of the nerolidyl cation, shown in gray, is not found in plant-derived sesquiterpenes. g-A = germacrene A, g-D = germacrene D, bcg = bicyclogermacrene.

cyclic cations can undergo further hydride shifts, methyl shifts, cyclizations, rearrangements, re- and de-protonations to form the final products of the enzyme [16]. Thus, the farnesyl and nerolidyl cations form the roots of a branching tree of hundreds of diverse intermediates and end products.

Many STSs are multi-product enzymes, with two of the more extreme examples being δ-selinene and γ-humulene synthases from *Abies grandis*, which produce 52 and 34 sesquiter-penes respectively. In order to determine whether cation specificity is maintained across minor products, we looked at the reaction pathways of the sesquiterpenes produced by the

multi-product enzymes in our previously assembled database [5]. In their review [17], Vattek-katte *et al.* looked into multi-product mono-, sesqui-, and triterpene synthases with respect to factors affecting their promiscuity, such as substrate isomers, metal cofactors and pH. However, they did not specifically address the similarity of an enzyme's minor products to the major product. The collation of characterized STSs in our database provides us with 96 multi-product STSs across a wide variety of species, to better analyze and address this question.

For each sesquiterpene, the route taken in the reaction tree, up to the depth shown in Fig 1 was determined as explained in Materials and Methods. Out of the 96 enzymes with more than one product, 79 (82%) had products from the same branch of the tree, three were 10,1-farnesyl synthases with products from different sub-branches, seven had products from the same cation but a different initial cyclization, and twelve synthases had products from different cations, including the aforementioned multi-product *Abies grandis* γ-humulene synthase. Of these twelve multi-cation STSs, however, eight had an acyclic farnesyl product in addition to neroli-dyl-derived compounds. The ease of formation of acyclic farnesyl products (a single step from the farnesyl cation) indicates that they can be formed even by a nerolidyl synthase as the farne-syl cation is the precursor of the nerolidyl cation. Thus there are only four examples of true multi-cation STSs (<5% of the experimentally characterized multi-product enzymes).

This analysis indicates that STSs are, in the vast majority of cases, optimized for the production of sesquiterpenes from a single, well-defined reaction route, by careful control of intermediates right from the commencement of the reaction, at the precursor cation formation step. This insight can be helpful in STS engineering: changing the reaction specificity of an existing STS to products in the same reaction pathway may be easier to accomplish, with fewer mutations, than the introduction of a new reaction pathway. For instance, the 412 active mutants made by O'Maille *et al.*, exploring the mutation space of tobacco 5-*epi*-aristolochene synthase and *Hyoscyamus muticus* vetispiradiene synthase, in many cases resulted in an increased production of germacrene A along with the original product 5-*epi*-aristolochene, which is derived from germacrene A [18]. Given that even multi-product STSs make sesquiterpenes from the same cation, understanding and predicting this cation specificity can greatly narrow down the possible products of a given enzyme.

## Structure-based cation prediction helps overcomes species bias

STS enzymes all have similar tertiary structures consisting entirely of α-helices and short connecting loops and turns. Each structure is typically organized into two domains, with the C-terminal domain containing the active site. The conserved nature of STS enzyme structures across the plant kingdom indicates that applying machine learning on attributes derived from these structures may explain more about cation and product specificity in STSs than sequence-derived attributes, which are more phylogeny-specific. However, due to the lack of available crystal structures for all the characterized enzymes, we turn to homology modelling to make up the deficit. Six crystal structures of STS enzymes were used for multi-template homology modelling of the C-terminal domains of 247 characterized plant STSs. Table 1 describes these six structures, three of which are farnesyl synthases, two nerolidyl synthases, and one is a cadalane-type synthase. S1 Appendix provides more detail on the modelling results, by comparing multi-template models to those created using the single closest template, and by comparing models of the six experimental structures to themselves. Models of the full enzyme sequences were also made but found to be sub-optimal due to the lack of a defined sequence alignment in regions surrounding the C-terminal domain. These results indicate that the final C-terminal domain models are accurate and capture the characteristics of the true structures in this region.

**Table 1. The six structures used for multi-template modelling.**

| Name | PDB ID | Resolution | Species | Product | Cation |
|------|--------|-----------|---------|---------|--------|
| GACS | 3G4F | 2.65Å | *Gossypium arboreum* | (+)-$\delta$-cadinene | cadalane |
| AGBS | 3SDU | 1.89Å | *Abies grandis* | $\alpha$-bisabolene | nerolidyl |
| AABS | 4FJQ | 2.00Å | *Artemisia annua* | $\alpha$-bisabolol | nerolidyl |
| AAHS | 4GAX | 1.99Å | *Artemisia annua* | $\gamma$-humulene | farnesyl |
| HMVS | 5JO7 | 2.15Å | *Hyoscyamus muticus* | vetispiradiene | farnesyl |
| TEAS | 5EAU | 2.15Å | *Nicotiana tabacum* | 5-*epi*-aristolochene | farnesyl |

In order to assess the effect of using features derived from modeled structures compared to purely sequence-based approaches we compared results across three classifiers. One is a simple rule-based classifier, Clf-id, that assigns a test sequence the same class as its closest training sequence based on sequence identity. While this approach is a good baseline and often used in biotechnological applications, machine learning-based models have two advantages over this simple model. Firstly, they are capable of incorporating more complex features, such as the sequence and structure features described in, as well as recognizing more complex patterns in these features, allowing for more accurate predictions that generalize across proteins. Secondly, trained machine learning models can be inspected to understand the patterns used for prediction [19, 20]. In this case, this can help gain insight into the contributions of different residues to cation specificity. Therefore, the other two classifiers use the hierarchical machine learning framework described in Materials and Methods with only sequence features (Clf-seq) and with sequence and structure features (Clf-str) respectively. Our classification frameworks make use of gradient boosting trees due to their good out-of-box performance and capability of handling missing feature values caused by deletions in some enzymes.

The dataset consists of 176 farnesyl cation-specific STSs and 72 nerolidyl cation-specific STSs. The remaining 25 STSs are not used for training as they either form products from both cations or only cadalane-type compounds. The cadalane skeleton (Fig 1) can be formed by either of the two precursor cations [21] or in acidic conditions of *in vitro* assays from rearrangements of germacrene D [22]. These two alternatives make it difficult to judge whether a cadalane STS goes through the farnesyl or the nerolidyl pathway.

Table 2 shows the performance of these three classifiers using increasingly difficult validation schemes: a random five-fold cross-validation (Random Split), a leave-10-genera-out based scheme (Genus Split), and, finally, training on 177 dicot STSs (124 farnesyl, 53 nerolidyl) with 48 monocot and coniferous STSs (29 farnesyl, 19 nerolidyl) in the test set (Clade Split). Due to

**Table 2. Validation results of three classifiers.**

| Scheme | Random Split | | | Genus Split | | | Clade Split | | |
|--------|------|-----|-------|------|-----|-------|------|-----|-------|
| Clf- | bAcc | AUC | AUPRC | bAcc | AUC | AUPRC | bAcc | AUC | AUPRC |
| id | 0.88 ± 0.05 | **0.88 ± 0.06** | 0.88 ± 0.05 | 0.72 ± 0.11 | 0.72 ± 0.11 | 0.69 ± 0.16 | 0.51 | 0.51 | 0.46 |
| seq | 0.88 ± 0.04 | 0.83 ± 0.05 | 0.94 ± 0.02 | 0.69 ± 0.07 | 0.88 ± 0.07 | 0.75 ± 0.16 | 0.51 | 0.62 | 0.54 |
| str | **0.90 ± 0.04** | 0.86 ± 0.03 | **0.94 ± 0.02** | **0.73 ± 0.07** | **0.89 ± 0.07** | **0.77 ± 0.13** | **0.64** | **0.75** | **0.59** |

1. Clf-id—sequence-identity rule-based classifier, 2. Clf-seq—classification framework using sequence features, 3. Clf-str—classification framework using sequence and structure features. Each column section shows the results of a different validation scheme: randomized 5-fold cross validation (Random Split), genus-based cross validation (Genus Split), and training on 177 dicot STSs and testing on 48 monocot and conifer STSs (Clade Split). For each scheme, balanced accuracy (bAcc), area under the ROC curve (AUC), and area under the precision-recall curve (AUPRC) are presented. The Random Split and Genus Split are repeated 5 and 10 times respectively, leading to the reported standard deviation values.

the imbalanced nature of the dataset, we use a variety of different metrics to measure performance. These are further described in the Materials and Methods. While Clf-str outperforms the sequence-based approaches by a small margin in the random cross-validation results, the improvement is much more striking in the phylogenetic validation schemes. As STS sequence similarity is biased more towards phylogeny than functional activity, Clf-id and Clf-seq make more errors when testing on species far away from those in the training set. Since Clf-str uses structure-derived information, it is less affected by this bias. This indicates that the structure-based classification framework is more suited to be applied across all plant species, including under-explored species, without losing out on predictive performance. S1 Fig shows the predicted nerolidyl percentages for each enzyme with Clf-str (using the probabilities returned by the genus-based split for each enzyme in the dataset). A clear separation is seen between farnesyl and nerolidyl-cation specific enzymes. However, because of the much lower number of nerolidyl-cation specific enzymes in our dataset, the nerolidyl predicted probabilities for nerolidyl-cation specific enzymes (average 53% ± 30%) are generally lower than the farnesyl predicted probabilities of farnesyl-cation specific enzymes (average 88% ± 19%, calculated as 100 —nerolidyl predicted probability percentage).

As a consequence of its superior performance, the structure-based classifier likely finds features and residues that are important for cation specificity across all plant species—something we can look into to understand generic STS cation determinants.

Thirty cation-specific residues were selected from Clf-str, as described in Materials and Methods. Fig 2 visualizes the characterized STS enzymes with respect to the features values of the cation-specific residues, colored by cation and cyclization specificity. Though imperfect, a separation of farnesyl and nerolidyl cation-specific STSs can be seen. Most cadalane STSs lie on the farnesyl side, with only two being predicted as nerolidyl cation-specific STSs in the Genus Split results. This can indicate that many cadalane synthases in fact make their products through a germacrene D intermediate, or, if the measurements were conducted *in vitro*, then perhaps acidic assay conditions led to spontaneous product rearrangements, thus the interpretation of Fig 2 in terms of STSs producing only cadalane products is unclear. While nerolidol synthases (N-acyclic in Figs 1 and 2) cluster separately from the rest, farnesene and farnesol synthases (F-acyclic in Figs 1 and 2) are found all across the reduced space. Due to the ease of formation of these acyclic farnesyl products, it is possible that ancestral versions of these enzymes did indeed produce nerolidyl-derived compounds but this capability was later lost.

A further test of Clf-str was performed on 42 STS enzymes characterized from August 2017-January 2020, not included in the first release of the characterized STS database [5], 31 of which come from species not present in the current set. This new set consists of 24 farnesyl cation-specific STSs, 16 nerolidyl cation-specific STSs, three STSs producing only cadalane compounds, and one STS which produces both farnesol and nerolidol. Clf-str correctly predicted all the nerolidyl cation-specific STSs and all but two of the farnesyl cation-specific STSs. Both the cadalane and the acyclic STSs were predicted as farnesyl cation-specific STSs. These enzymes are listed in S1 Table and have been added to the second version of the characterized STS database, found at bioinformatics.nl/sesquiterpene/synthasedb.

## Residues in five structural regions contribute to cation specificity

The cation-specific residues according to our structure-based predictor are indicated in Fig 3A on the tobacco *epi*-aristolochene synthase (TEAS) structure. They are roughly found in five different structural regions, labeled A-E. Also shown are the residues in the three known terpene synthase motifs, namely RxR, DDxxD, and NSE/DTE, as well as the magnesium ions and substrate analog. Fig 3B shows the sequence composition of these thirty residues across farnesy

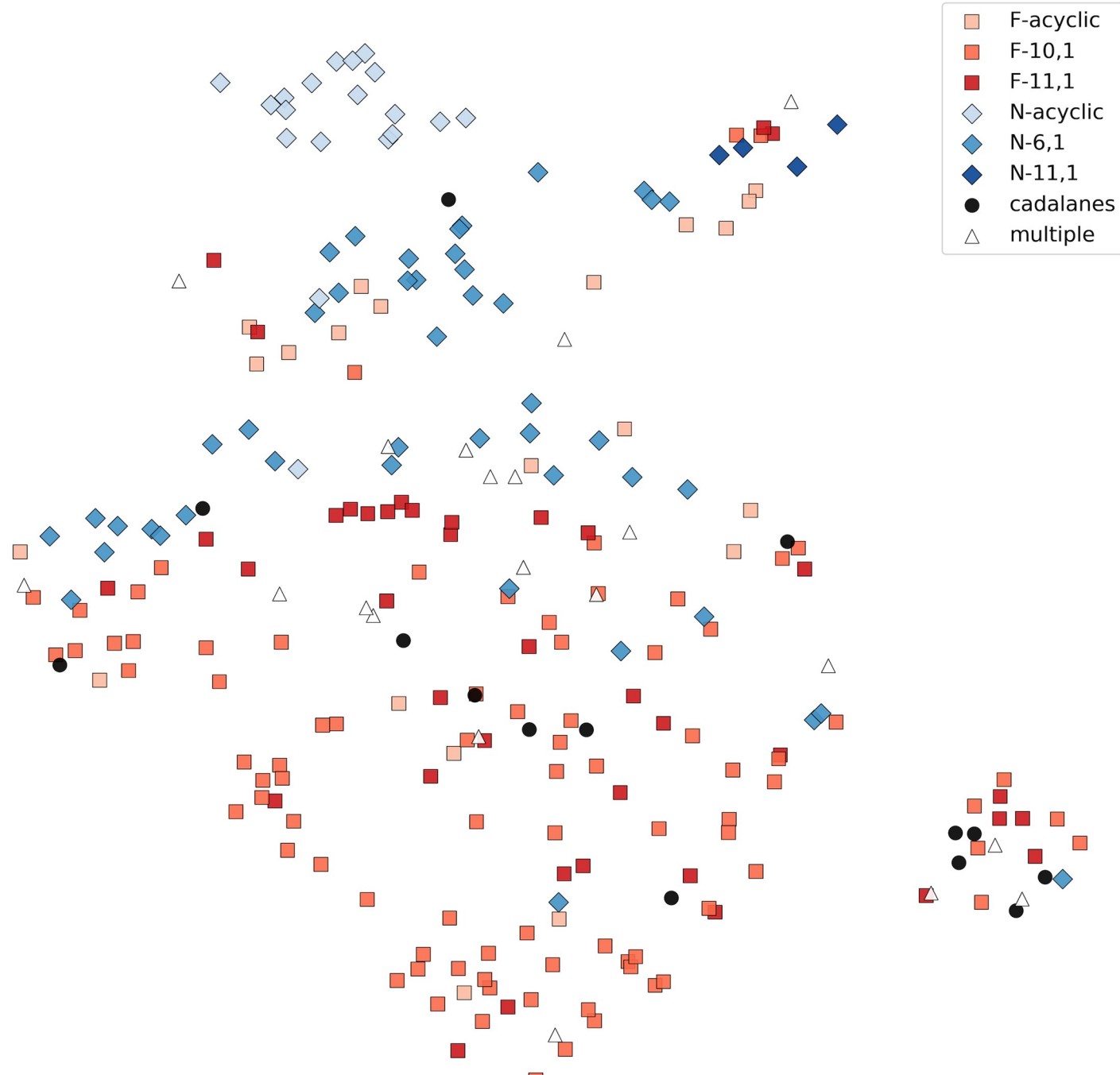

**Fig 2. Characterized STSs visualized using the feature values of the cation-specific residues followed by dimensionality reduction using UMAP [23], which positions STSs with similar feature values closer to each other.** Squares represent farnesyl cation-specific STSs and diamonds represent nerolidyl cation-specific STSs. Each STS is also colored by its cyclization specificity. Enzymes catalyzing products from different precursor cations are marked as triangles.

and nerolidyl cation-specific STSs. While the sequence logos (Fig 3B) show significant differences in some predictive positions, others have very similar amino acid distributions across the two cations, indicating that their differences lie solely in some combination of their structural features likely due to their structural interaction with neighboring residues. Thus, these

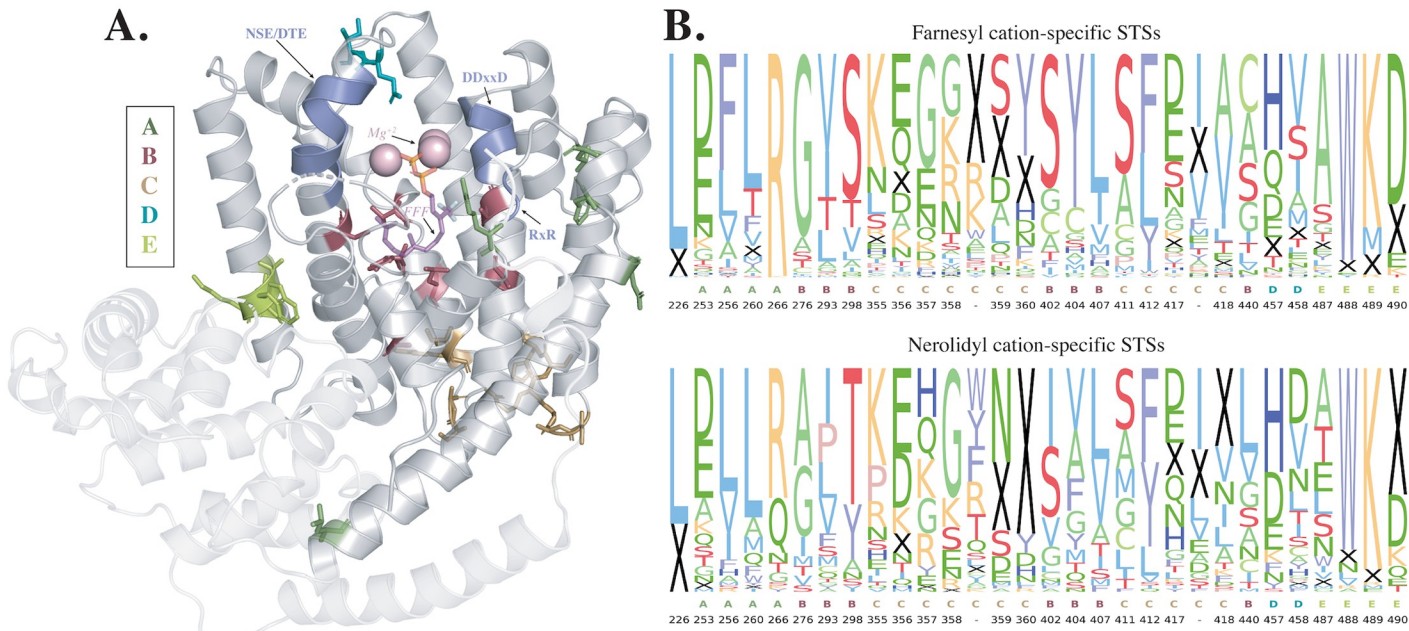

**Fig 3. A**. Thirty cation-specific residues found by the structure-based Clf-str predictor on the tobacco *epi*-aristolochene synthase (TEAS) structure, colored by region. Terpene synthase motifs are labelled in purple, $Mg^{+2}$ ions in pink, and a substrate analog in orange and dark pink. The N-terminal domain is shaded with a lower opacity. **B**. Sequence conservation of Clf-str cation-specific residues across farnesyl and nerolidyl cation-specific STSs, labelled by region and residue position in the TEAS structure. The height of a letter represents its frequency in that position. An insertion/deletion is represented by a black 'X'. Residue positions which are deleted in the TEAS structure are represented by '-'s and correspond to residue 627 and residue 687 respectively in the *Abies grandis α*-bisabolene synthase structure (PDB ID: 3SDU). Note that if, for a given position, the residues in both logos are similar, this indicates that in spite of similarity in sequence at this position, the farnesyl and nerolidyl cation-specific enzymes are structurally different.

residues would not have been identifiable from sequence-based analysis alone, further demonstrating the power of the integrative approach presented here. S2 Fig shows residue scores across the 10 folds in the genus-based split. The scoring is consistent irrespective of the training set used, indicating that these residues are indeed catalytically important across all plant species.

To obtain more information about these thirty residues we turned to the wealth of uncharacterized putative terpene synthase enzymes in sequenced plant genomes and transcriptomes. The products of these putative enzymes are unknown, so they cannot be used to train a classifier; however the sequences themselves still carry valuable information about conservation and divergence. We used co-evolutionary analysis to inspect these sequences in the context of the cation-specific residues. Co-evolutionary analysis is a statistical technique applied on protein sequence alignments based on the underlying biological theory of residue co-evolution [24]. This theory postulates that if there is a mutation in one residue involved in an interaction, then proteins in which its interaction partner is mutated as well, in a way that maintains their interaction, are preferentially selected by evolution. While this technique is most often used to find potentially interacting residues within a protein in protein families with scant structural information, an alternative scenario of co-evolution can play out in the case of functionally related residues [25]. For instance, two residues which contact a substrate or an intermediate, while not interacting directly, may still co-evolve to maintain their shared interactions with the substrate.

We used 8344 putative terpene synthase N- and C-terminal domains obtained from sequenced plant genomes and transcriptomes to perform a co-evolutionary analysis as described in

Materials and Methods. S3B and S3C Fig show the predicted contact map from this analysis compared to the pairwise minimum $\beta$-carbon Euclidean distance matrix across the six structures in Table 1. When looking at the top 1500 predicted contacts (S3A Fig), 328 have residues at least 7 positions apart in the sequence indicating long range interactions across different structural regions. Only 78 (24%) of these are not capable of physical interaction (>11 Å apart) in all of the six STS crystal structures. 10 of these predicted pairs, shown in Fig 4, have at least one residue among the thirty cation-specific residues. Below, we discuss specific examples of these residues and pairs in context of the five regions predicted to be involved in cation specificity.

Residues in region A (colored dark green in Figs 3 and 4) lie in the A-C loop, close to the conserved RxR motif, with one residue forming the second Arg in the motif itself. This motif has been implicated in the complexation of the diphosphate moiety, preventing nucleophilic attacks on any of the intermediate carbocations [26]. As this is one of the first steps to occur in order for the resulting charged intermediate to undergo cyclization and further reactions, it can play a crucial role in determining how the newly formed cation is positioned, thereby determining whether a farnesyl cation is formed or a nerolidyl cation. In previous work we showed that many nerolidol (N-acyclic) synthases have a mutation in this motif, from RxR to RxQ (as can be seen in the sequence logo; Fig 3B, position 266), indicating that changes in and around this motif can indeed affect the products formed.

The six residues in region B (colored red in Figs 3 and 4) all lie right in the center of the active site cavity, in helix D (G276, T293, S298, in TEAS), around the kink region in helix G2 (T402, Y404, L407) and in helix H2 (C440), enveloping the descending substrate from all sides. The residues in this region are very close to both the substrate analog co-crystallized with TEAS as well as the analog co-crystallised with *Abies grandis* $\alpha$-bisabolene synthase, as depicted in Fig 4C. This proximity has led to a more thorough exploration of these residues in the context of product specificity, than in other regions of the structure. For instance, Yoshikuni *et al*, 2006 explored plasticity residues in the active site of the promiscuous *Abies grandis* $\gamma$-humulene synthase [8]. Among the many mutants they made, those that converted the major product from the farnesyl-derived $\gamma$-humulene to nerolidyl-derived products such as $\beta$-bisabolene, $\alpha$-longipinene, longifolene, and sibirene, contained mutations in the residues corresponding to T402, Y404 and C440 in TEAS—three cation-specific residues according to our predictor. Two of these residues (Y404 and C440) have also been explored by Salmon *et al* [27] when mutating the acyclic $\beta$-farnesene synthase from *Artemesia annua* to a cyclic nerolidyl cation-derived enzyme.

Similarly, Li *et al*, 2013 demonstrated that a single mutation in the kink in the G2 helix can change the product specificity of an *Artemisia annua* STS from $\alpha$-bisabolol, a nerolidyl-derived sesquiterpene, to the farnesyl-derived $\gamma$-humulene [28]. T402 from this kink has co-evolved with S298 in the parallel helix D. As depicted in Fig 4B (column 1), while these two positions are very often both Serine in farnesyl cation-specific STSs, in nerolidyl cation-specific STSs the commonly occurring pairs are Thr-Ile or Tyr-Ser. The dipole of T402 has been implicated along with T401 in directing the cationic end of the farnesyl chain into the active site, preparing it for a C10 attack [26]. Isoleucine, which is not often found to be a catalytic residue due to its inert nature, cannot perform this task in nerolidyl cation-specific STSs. Another contact is between the cation-specific residue C440 and Y376 (numbered 2 in Fig 4B). A mutational analysis on a multi-product maize STS by Kollner *et al.* demonstrated the importance of Y376 in the formation of bicyclic products such as sesquithujene and bergamotene, derived from the nerolidyl cation [29]. The residue positioned three residues downstream of Y376 was identified by Kollner *et al* in 2009 to be involved in controlling the ratio of $\alpha$-bergamotene to the acyclic $\beta$-farnesene in maize STS orthologs [30]. Therefore, the combined effects of position 376 and

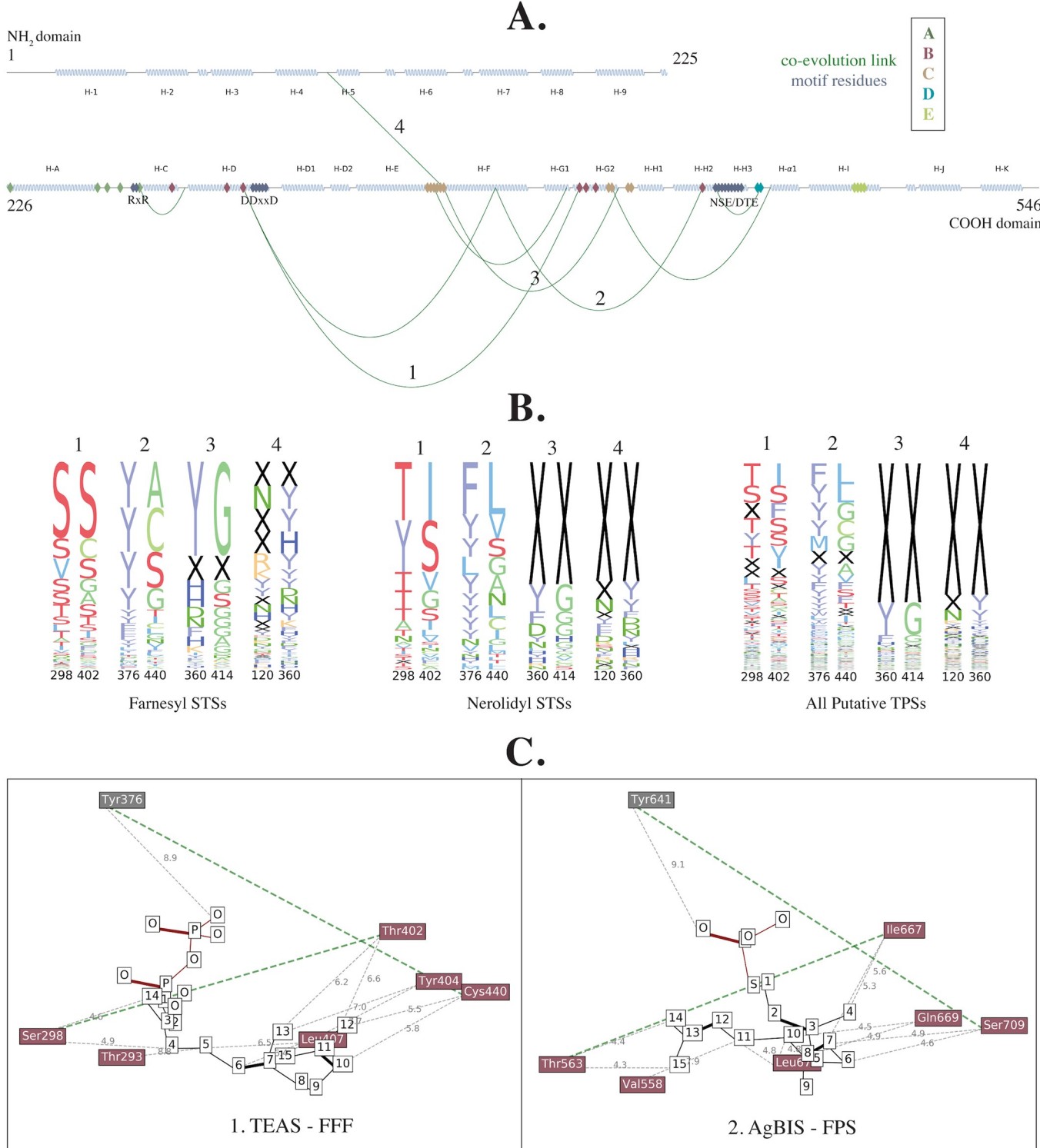

**Fig 4. A**. Tobacco *epi*-aristolochene synthase (TEAS) secondary structure with distal cation-specific co-evolutionary contacts (green arcs), motif residues (purple), and cation-specific residues (colored by region). Helix naming as in Starks *et al.* [26] **B**. Sequence-pair conservation of four cation-specific contacts discussed in the text, across farnesyl and nerolidyl cation-specific STSs, and all putative terpene synthases. The height of a pair of letters represents the frequency of the pair appearing in those two positions, with 'X' representing gaps. **C**. Diagrams indicating the proximity of residues labeled B in Fig 3B, as well as the residues that they co-evolve with, to substrate analogs trifluorofarnesyl diphosphate (FFF) co-crystallized with TEAS (left) and farnesyl thiodiphosphate (FPS) co-crystallized with *Abies grandis* α-bisabolene synthase (AgBIS) (right). Carbon atoms are numbered (white boxes) as in the FFF substrate analog moiety in PDB

ID 5EAU. The closest distance (in Å) between each residue's $\beta$-carbon and a substrate atom is labeled in gray. Two co-evolving contacts (labeled 1 and 2 in **A**) are colored in green.

440 are likely required for the formation of the nerolidyl cation followed by a second cyclization to bicyclic nerolidyl sesquiterpenes. An alignment of TEAS with the examples discussed here is depicted in S4 Fig. These examples demonstrate that residues found important by our structure-based predictor are indeed involved in catalytic and functional activity. They also establish the power of an integrative machine learning approach to pinpoint residue positions important across a variety of species, a combination of what one would find from each of the individual studies referenced above. A Fisher's exact test for the significance of the number of residues found both by our predictor and in literature returned a *p*-value of $9.8e^{-07}$.

The 12 residues in region C (colored orange in Figs 3 and 4) encompass the entire E-F loop and parts of the G2-H1 loop at the very bottom of the active site cavity. An interesting residue here is H360, the last residue in the E-F loop. Sequence conservation shows that this position is very often deleted in nerolidyl cation-specific synthases, while farnesyl cation-specific synthases usually have bulky residues such as Tyrosine and Histidine (Fig 3B, position 360). Two of its co-evolving partners (numbered 3 and 4 in Fig 4B), one from the parallel helix G2 and one from the 4-5 loop in the N-terminal region, are also primarily deleted in nerolidyl cation-specific STSs but present in farnesyl cation-specific STSs, albeit usually as Glycine in helix G2. While the connection with the N-terminal domain is surprising, the parallel residue in the C-terminal domain, when present, may physically interact at some point during the reaction or in other plant STSs, not captured in the six crystal structures currently available [31]. A deletion can break this interaction, which in turn can have an effect on the positioning of helix G2 in the active site and thereby the positioning of the cation-specific residues that lie within it. These subtle alterations in cavity shape may in turn affect which kinds of intermediates fit comfortably inside the cavity.

Two consecutive high scoring residues (region D, colored blue in Figs 3 and 4), lie in the H3-$\alpha$1 loop, close to the catalytic NSE/DTE motif. This motif is involved in coordinating $Mg^{+2}$ ions along with the DDxxD motif on the opposite side [32]. This region lies at the entrance of the active site cavity and is in an optimum position to contact the substrate as it enters the cavity. In addition, the inability to crystallize this region in three of the six crystal structures indicates that this loop is very flexible [33].

Residues in region E (colored light green in Figs 3 and 4) lie in helix I, near the end of the C-terminal domain and close to helix 7 and helix 8 in the N-terminal domain.

Overall, these results show that cation-specific residues in regions labelled A, B, and D lie within areas known to participate directly in the catalytic reaction. These residues were predicted by our machine learning approach without using any knowledge on their functional properties. Some of these residues have been mutated before and were shown to be important for cation specificity. This indicates that the other residues are also likely to perform similarly crucial roles, perhaps also in STSs that have not been used so far in mutagenesis experiments. Residues labeled C and E lie quite far from the active site and could be involved in subtle alterations of the cavity shape or in stabilising contacts with the N-terminal domain. Though this domain is known to be important for plant STS reactions, its exact function has not been fully explored. However, just as O'Maille *et al.* showed that residues distant from the active site can still be functionally crucial [18], these distal residues are likely to have multifaceted and interdependent roles in cation specificity that only such large-scale computational approaches can recognize. Further experiments and mutational studies in these regions are required to confirm and elaborate their involvement in the STS reaction mechanism. Meanwhile, the

structure-based predictor, as well as the cation-specific sequence and contact conservation information described can be used to screen through the many thousands of uncharacterized putative STSs with a particular cation specificity in mind as demonstrated in the next section.

## Bisabolyl cation synthases from *Citrus bergamia* 'Femminello'

One potential application of the cation-specificity predictor presented here is to screen for enzymes with a desired specificity. We demonstrate this application to find STSs catalyzing the formation of products derived from the bisabolyl cation from 23 terpene synthase-like sequences extracted from the transcriptome of *Citrus bergamia* 'Femminello' (described in Materials and methods). Using the hidden Markov model approach detailed by [6], 11 sequences out of these 23 were predicted to be STSs (as opposed to mono- or diterpene synthases). We used the cation specificity predictor on these 11 and sorted by decreasing order of predicted nerolidyl cation specificity, selecting enzymes with predicted probability percentage above 10%, based on the predicted percentages of the characterized database (S1 Fig).

Two enzymes clustered close to the nerolidol cluster in Fig 2 and were thus excluded, resulting in four enzymes with >10% predicted nerolidyl cation specificity. Three of these could be experimentally characterized, submitted to GenBank with identifiers MT636927, MT636928 and MW384854 respectively. MT636927 and MT636928 produced bisabolyl cation-derived products. MT636927 has 55% predicted nerolidyl specificity and produced trans-$\alpha$-bergamotene, $\beta$ bisabolene, and $\alpha$ bisabolol. MT636928 has 11% predicted nerolidyl specificity, and produced zingiberene. MW384854 has 26% predicted nerolidyl specificity but produced the farnesyl-cation derived caryophyllene. The chromatograms and the fragmentation patterns of the identified peaks and the reference compounds can be found in S5 Fig and S2 Appendix.

Sequence identity based screening, on the other hand, predicts all 11 enzymes as farnesyl cation specific showing that based on only sequence identity, we cannot prioritize candidate genes for production of bisabolyl cation-derived products. Thus, the cation specificity predictor can be used for effective screening of STSs with desired intermediate specificity, saving time, labour and costs required for extensive experimental characterization. Considering that the bisabolyl cation is one of the least represented intermediates in our dataset, expanding the number of experimentally characterized enzymes used for training can further increase the accuracy of our results, and even allow for more fine-grained product specificity prediction.

## Conclusion

The availability of growing numbers of characterized and putative sesquiterpene synthases opens doors for the application of computational analyses in order to obtain insights about this large and amazingly diverse family of enzymes. While STSs collectively produce many hundreds of compounds, these are all rearrangements of two precursor carbocations deriving from a single substrate. We show that multiproduct STS enzymes catalyze the formation of products deriving from the same cation, indicating that cation specificity is determined early in the reaction. A combination of structure-based supervised machine learning and unsupervised co-evolution gives us a set of structural regions implicated in cation specificity determination as well as possible functional relationships between residues in these regions and other parts of the STS structure. The predictor itself can be used for cation-specificity screening, while the residues and corresponding linkages discussed here can be used to design mutational studies with a higher likelihood of maintaining catalytic activity while changing cation specificity. Such an integrative approach can also be applied to other diverse enzyme families in order to uncover large-scale interdependent relationships between catalytic residues influencing product specificity. As the number of characterized STSs from across the plant kingdom

increases, more specific predictors can be designed, in order to screen STSs at the cyclization or even product level.

# Materials and methods

## Reaction pathway determination

The reaction pathway for each sesquiterpene in the database was determined using the scheme detailed in IUBMB's *Enzyme Nomenclature* Supplement 24 (2018) [34] up to the depth specified in Fig 1. For example, the sesquiterpene viridiflorene would be labeled F112 as it derives from bicyclogermacrene which itself is labeled F11. Sesquiterpenes derived from the cadalane skeleton, namely cadinanes, cubebenes, copaenes, amorphenes, sativenes, muurolenes, ylangenes, and their alcoholic variants, are marked as cadalanes as they can form from multiple reaction pathways.

Two sesquiterpenes share a reaction path if the pathway annotation of one is a non-strict prefix of the other's. For example, sesquiterpenes labeled F1, F11, and F113 belong on the same reaction path while those labelled F111, F112, and F12 do not. If multiple cadalane-type compounds are produced by one enzyme, they are assumed to come from the same path. These rules are used to calculate the number of multi-product enzymes with products following the same reaction path.

STSs were labelled as farnesyl or nerolidyl according to the group that their products belong to. STSs making cadalane products along with additional non-cadalane products are labeled with the cation of these other products. Multi-product STSs producing compounds from different cations, as well as cadalane STSs without any non-cadalane product are considered separately and are not used for training.

## Sequence extraction and alignment

N-terminal and C-terminal domain sequences were extracted from all spermatophyte plant STSs from the database using HMMER [35] and the Pfam [36] domains PF01397 and PF03936 respectively. All N-terminal and C-terminal sequence alignments were made using Clustal Omega [37], using the corresponding Pfam domain HMM to guide the alignment. A combined N- and C-terminal domain HMM was built by aligning each half of the common seed sequences from both respective Pfam domains, stacking the resulting alignments together, and using the hmmbuild tool in HMMER [35]. This HMM is referred to as Terpene_synth_N_C.

## Homology modelling

For each STS, 500 multi-template homology models were created of the C-terminal domain region using MODELLER [38], with six STS structures from the PDB [39] as templates, as listed in Table 1. These were aligned to each sequence using the C-terminal PF03936 Pfam domain [36] as a guide, using Clustal Omega [37]. The top three models were selected based on their N-DOPE score for feature extraction.

For comparison, 500 models were also made using a single template for each enzyme; the template chosen was the one having the maximum sequence identity to the enzyme being modelled. Similarly, models were made for each of the six template structures using the other five structures as templates. Models of full STS sequences (including the N-terminal domain) were also made using a similar multi-template approach with the custom Terpene_synth_N_C HMM to guide the alignment to the templates. Results for these three additional approaches are presented in S1 Appendix.

## Feature extraction

Sequence and structure features were extracted from each STS as described below and aligned according to the C-terminal domain alignment. Gaps in the alignment were represented as NaNs for continuous features and as a separate category for categorical features.

**Sequence features.**   For each STS sequence, PSIBLAST [40] was run on the non-redundant protein database (nr) [41] and used to calculate a position-specific scoring matrix (PSSM) and a position-specific frequency matrix (PSFM). The information content of each column in the PSSM was also calculated. SCRATCH [42] was used to predict the secondary structure and surface accessibility of each residue. Finally, the raw amino acid sequence was also used as a feature source. Categorical features were one-hot encoded.

**Structure features.**   Structural features were extracted for each of the top three homology models for each STS. All atom-level features were converted into $\alpha$-carbon, $\beta$-carbon, and mean residue features. For Gly, the $\alpha$-carbon was used for the $\beta$-carbon features as well. ProDy [43] was used to calculate the 50-mode Gaussian Network Model (GNM) and Anisotropic Network Model (ANM) atom fluctuations using the calcGNM/calcANM functions followed by the calcSqFlucts function. APBS [44] was used to calculate the Coulomb and Born electrostatics of a modelled structure. PDB2PQR [44] was first used to generate a PQR file from each PDB file, followed by running the born command with an epsilon (solvent dielectric constant) of 80 and the coulomb command with the -e option. DSSP features are calculated using ProDy [43] to give hydrogen bond energies, surface accessibility, dihedral angles ($\alpha$), bend angles ($\kappa$), $\phi$, and $\psi$ backbone torsion angles, and tco angles (cosine angle between the C = O of residue $i$ and the C = O of residue $i - 1$). Residue depths were extracted using BioPython [45] from the PDB files of the top three models.

## Classification framework

A classification framework using Gradient boosting trees (as depicted in S6 Fig) was built for different sets of features. The framework is trained in three steps:

1. A separate gradient boosting tree is trained for each kind of feature for all residues. XGBoost [46] was used with default parameter settings for these intermediate classifiers (100 trees, learning rate = 0.1, gamma = 0, subsample = 1, colsample_bytree = 1, colsample_bylevel = 1). These simple settings are sufficient as these classifiers are only used to find predictive residues, as described in the next step.

2. The sum of normalized weights for each residue across all the trained feature models from Step 1 is used as a scoring measure to select the top thirty residues.

3. A final gradient boosting forest with much stricter parameter settings (2000 trees, learning rate = 0.005, gamma = 0.01, subsample = 0.7, colsample_bytree = 0.1, colsample_bylevel = 0.1) is trained using XGBoost [46] on all the feature values of the top residues picked in Step 2. These parameter settings are chosen to make a more conservative classifier that avoids overfitting in three ways: reduced model complexity by regularization (using the gamma parameter), robustness to noise by random selection in each intermediate tree of both data points (the subsample parameter) and features (the colsample parameters), and a slow learning rate combined with a large number of trees to increase the power of the ensemble.

For testing, the features of the selected thirty residue positions in the test enzymes are fed into the trained classifier.

Clf-seq and Clf-str are two classifiers built using this framework utilizing only sequence features and both sequence and structure features, respectively. Clf-id is a simple rule-based classifier that does not use this framework and instead returns the class of the closest training set sequence based on sequence identity.

## Validation and testing

Three validation schemes are used to test a classifier.

1. Random Split: A random five-fold cross-validation with 80%-20% train-test split.

2. Genus Split: A scheme in which cases from 65 genera are used for training and the rest for testing, repeated 10 times with different sets.

3. Clade Split: All dicot STSs are used for training and monocot and conifer STSs for testing.

Three different metrics are used to measure the performance of each classifier, using the definitions of $TP$ and $TN$ as the number of nerolidyl cation-specific synthases and number of farnesyl cation-specific synthases predicted correctly at a certain threshold of predicted probability, and $FP$ and $FN$ as the number of nerolidyl cation-specific synthases and number of farnesyl cation-specific synthases predicted incorrectly at a certain threshold. All metrics are calculated using the scikit-learn Python library [47].

1. Balanced accuracy (bAcc): $\frac{1}{2}\left(\frac{TP}{TP+FN} + \frac{TN}{TN+FP}\right)$ at a threshold of 0.5.

2. Area Under the Receiver Operating Characteristic Curve (AUC): Calculated as the area under the plot of the fraction of $TP$ out of the total number of nerolidyl cation-specific synthases vs. the fraction of $FP$ out of the total number of farnesyl cation-specific synthases, at various threshold settings.

3. Area Under the Precision-Recall Curve (AUPRC): Calculated as the area under the plot of the precision ($TP/(TP + FP)$) vs. the recall ($TP/(TP + FN)$) at various threshold settings.

42 newly characterized synthases from literature (listed in S1 Table) are used as the final independent test set.

## Selecting cation-specific residues

The normalized weights across all feature classifiers were summed across all the folds of the Genus Split and the resulting thirty highest scoring positions represent the set of cation-specific residues. The sequence and structural features of these residues were used to visualize the set of characterized STSs. This was done by applying UMAP [23] to reduce the dimensionality to 2.

## Co-evolution analysis on plant terpene synthase-like proteins

An HMM search was performed using HMMER [35] and the custom Terpene_synth_N_C HMM across all plant UniProt proteins [48] and all plant transcriptome sequences from the OneKP transcriptome dataset [49]. Only those with sequence length at least one standard deviation away from the mean sequence length of the characterized STSs from the database [5] were retained. The resulting set of uncharacterized sequences were aligned with Clustal Omega [37] using the same HMM and 10 guide-tree/HMM iterations (clustalo option –iter = 10). Alignment positions not present in any of the six structures in Table 1 were discarded.

CCMPred [50] was used to perform co-evolution analysis on this alignment. The top 1500 predicted contacts were selected based on their confidence scores (S3A Fig). Contacts containing one residue from the cation-specific positions, at least 11 Å apart in any of the six structures in Table 1 and seven residues apart in sequence were retained.

### Visualization of cation-specific residues and contacts

Cation-specific residues and contacts were visualized in multiple ways.

- **3D Structure**—Pymol [51] was used to visualize the three-dimensional structure of tobacco 5-epi-aristolochene synthase (TEAS, PDB ID: 5EAU) and label terpene synthase motif residues and cation specific residues.

- **Sequence and Co-evolution Conservation Logos**—The positions of predictive residues in farnesyl and nerolidyl cation-specific STSs were used to generate two sequence conservation logos based on the percentage of appearance of each amino acid at each position. The sequence conservation of four co-evolving residue pairs was also visualized across farnesyl and nerolidyl cation-specific STSs and the set of putative terpene synthases. These figures were made with matplotlib [52].

- **Co-evolutionary Links**—The cation-specific residues and contacts as well as terpene synthase motif residues were visualized on the secondary structure of the N-terminal and C-terminal domain portions of the tobacco aristolochene synthase (TEAS) structure found by the two respective Pfam domains (PF01397 and PF03936), using matplotlib [52]. Helices are labeled as described by Starks *et al.* [26].

- **Substrate Analog Proximity**—Substrate analogs trifluorofarnesyl diphosphate (FFF) and farnesyl thiodiphosphate (FPS) were extracted from tobacco *epi*-aristolochene synthase PDB ID: 5EAU, and *Abies grandis* α-bisabolene synthase PDB ID: 3SAE respectively. Their positions in both structures were obtained by superposing the two structures to each other using the align command in Pymol [51]. Distances between a subset of the cation-specific residues and the atoms of the substrate analogs were visualized using matplotlib [52]. The atoms in both analogs are numbered according to the numbering of FFF.

### *Citrus bergamia* 'Femminello' STSs

The cation specificity predictor was employed to select four STSs among the putative terpenes synthases from *C. bergamia* with the highest nerolidyl cation specificity. The sequences were codon optimised, synthesised and expressed in *Rhodobacter sphaeroides*, as described earlier in [53]. The analysis of the products coming from the engineered strains was performed on the GC Agilent 7890B coupled to the MS Agilent 5977B. The used column is an HP-5MS 30m x 250um x 0.25um. The resulting chromatograms and the fragmentation patterns of the identified peaks and the reference compounds can be found in S5 Fig and S2 Appendix.

### Supporting information

**S1 Appendix. Homology modelling.** Results from different homology modelling runs. (PDF)

**S2 Appendix. Fragmentation patterns.** Fragmentation patterns of identified peaks from chromatograms and of corresponding reference compounds for *Citrus bergamia* STSs. (PDF)

**S1 Table. 42 STSs characterized from August 2017-January 2020, used as an independent test set.**
(TXT)

**S1 Fig. Predicted nerolidyl percentages.** Nerolidyl prediction percentages returned by Clf-str on characterized STSs calculated using the genus-based split.
(PDF)

**S2 Fig. Residue importance scores.** Residue scores found by Clf-str across 10 different train-test splits based on genus.
(PDF)

**S3 Fig. Predicted contacts from co-evolutionary analysis. A**. Scores of predicted contacts from co-evolutionary analysis in decreasing order. The 1500 contacts on the left of the orange dashed line are considered in the text. **b**. Pairwise minimum $\beta$-carbon distance matrix (in Å) across all six template structures in Table 1 for the residue positions present in the tobacco aristolochene synthase (TEAS) structure. **c**. The top 1500 predicted co-evolving contacts on the TEAS structure, indicated in black.
(PDF)

**S4 Fig. Alignment of discussed STSs.** Sequence alignment of tobacco aristolochene synthase with STS examples discussed in text.
(PDF)

**S5 Fig. *Citrus bergamia* STS chromatograms.** Chromatograms obtained from the *R. sphaeroides* strain expressing **a**. MT636927, **b**. MT636928, and **c**. MW384854 with peaks labelled.
(PDF)

**S6 Fig. Classification framework.** Classification framework of gradient boosting trees, to deal with a large number of residue-based features as well as allow for intuitive selection of the most predictive residues. A separate gradient boosting tree (GbT-s) is trained on each kind of feature across all aligned residue indices. The weights obtained from all the GbT-s classifiers are pooled (by taking the sum of the normalized weights) to select the top 30 residues. All feature values of these selected residues are used as input to the final classifier GbT. The parameter settings of the GbT-s and GbT classifiers are listed in Materials and Methods.
(PDF)

## Acknowledgments

We thank Miguel Correa Marrero for valuable feedback on early drafts and stimulating discussions on the approaches discussed in the text. We also thank Kenneth Paul Rivadeneira Guadamud for collecting 21 of the recently characterized sequences used in the test set.

## Author Contributions

**Conceptualization:** Janani Durairaj, Jules Beekwilder, Aalt D. J. van Dijk.

**Data curation:** Janani Durairaj.

**Formal analysis:** Janani Durairaj.

**Funding acquisition:** Harro J. Bouwmeester, Jules Beekwilder, Dick de Ridder, Aalt D. J. van Dijk.

**Investigation:** Elena Melillo.

**Methodology:** Janani Durairaj.

**Project administration:** Harro J. Bouwmeester, Jules Beekwilder, Dick de Ridder, Aalt D. J. van Dijk.

**Software:** Janani Durairaj.

**Supervision:** Harro J. Bouwmeester, Jules Beekwilder, Dick de Ridder, Aalt D. J. van Dijk.

**Validation:** Elena Melillo.

**Visualization:** Janani Durairaj, Elena Melillo.

**Writing – original draft:** Janani Durairaj.

**Writing – review & editing:** Elena Melillo, Harro J. Bouwmeester, Jules Beekwilder, Dick de Ridder, Aalt D. J. van Dijk.

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
