## [Decision Letter · Decision Letter 0]

29 Sep 2020

Dear Ms. Durairaj,

Thank you very much for submitting your manuscript "Integrating structure-based machine learning and co-evolution to investigate specificity in plant sesquiterpene synthases" for consideration at PLOS Computational Biology.

As with all papers reviewed by the journal, your manuscript was reviewed by members of the editorial board and by several independent reviewers. In light of the reviews (below this email), we would like to invite the resubmission of a significantly-revised version that takes into account the reviewers' comments.

We cannot make any decision about publication until we have seen the revised manuscript and your response to the reviewers' comments. Your revised manuscript is also likely to be sent to reviewers for further evaluation.

Sincerely,

Greg Tucker-Kellogg, PhD

Associate Editor

PLOS Computational Biology

Arne Elofsson

Deputy Editor

PLOS Computational Biology

Reviewer's Responses to Questions

**Comments to the Authors:**

Reviewer #1: This manuscript describes a computational approach to predict the carbocationic intermediate specificity of plant sesquiterpene synthases. While several previous studies have investigated residues responsible for product specificity by using mutant based approaches, the novelty of this work is a combined approach of homology modeling, machine learning, and co-evolutionary analysis. The authors take advantage of a large number of already characterized sesquiterpene synthases to define residues that contribute to cation intermediate specificity. While the approach does not predict specific sesquiterpene product outcomes it appears useful for facilitating screenings of terpene synthases with products derived predominantly from a farnesyl or nerolidyl cation intermediate. This type of analysis could be useful in similar predictions for other C10 or C20- terpene synthases.

Overall, the manuscript is well written and presented and I would like to add only minor comments and concerns.

1. The specificity predictions for uncharacterized sesquiterpene synthases from Citrus were between 10 and 50%. It would be advantageous to compare these results in terms of their robustness with predictions for enzymes that make products predominantly from farnesyl cations.

2. Salmon et al. (2015) described a residue network required for a transition from a linear to cyclic sesquiterpene product. Although this study followed a different goal, I am wondering whether the authors found and overlap in the residues that were investigated.

3. Fig. 1 It should be (E,E)-alpha-farnesene. FPP should be labeled (E,E)-FPP.

4. What do the numbers in white boxes represent in Fig. 4c? Carbon atoms?

5. Table S1 lists four beta-caryophyllene synthases from Acanthopanax sieboldianus. I am surprised that the product outcome of four enzymes is so similar; or do these enzymes make mixtures of products with beta-caryophyllene as major compounds?

6. The authors use the terms farnesyl and nerolidyl synthases. Since “farnesyl” and “nerolidyl” are not end products of these enzymes, I would better use the terms farnesyl cation- or nerolidyl cation-specific synthases.

7. Line 237: …show the predicted contact…

8. Line 350: “STSs with desired product specificity” is not absolutely correct since the analysis predicts intermediate specificity but not the final product profile.

Reviewer #2: This manuscript describes an interesting but clearly underdeveloped approach to computational analysis for functional assignment in the large sub-family of plant sesquiterpene synthases (STSs). The STSs represent an intriguing target for such analysis, with thousands of putative members, but only a couple of hundred that have been functionally (biochemically) characterized. Those characterized were previously assembled into a database by the authors, leading to the finding that the multiple products often observed in this sub-family still largely result from a single initial conformation of their common substrate, farnesyl diphosphate (FPP). Thus, even these promiscuous STSs generally exhibit some specificity in substrate folding/conformation within their active sites. Here the authors build on this finding. In particular, this pre-catalytic conformation directs the ensuing carbocation into certain reaction ‘channels’, first splitting between immediate cyclization (or terminating deprotonation) of the initially formed farnesyl carbocation versus isomerization to the nerolidyl carbocation that has additional cyclization options, with each of the subsequent cyclization options enumerated to allow assignment of reaction pathways (e.g., farnesyl is F1, nerolidyl is N1, with 10,1-cyclization of F1 then assigned as F11, while that of N1 as N11). This was coupled with protein modeling to identify thirty residues that correlate with pathway/carbocation. However, it is unclear why thirty were chosen. Examination of the sequence logos for these thirty for the first level of prediction (F1 vs N1), indicates that many of these positions do not seem significantly different (226, 253 & 260), while others do (e.g., 293 and 298). The authors do discuss previously reported mutational analyses that seem to support the relevance of some of these residues, but these are obviously selected examples. It would be much more convincing if the results were used in predictive fashion and then tested. For example, such analysis suggests that most of the cadalane producing STSs actually proceed via farnesyl rather than nerolidyl, although this would then require several more steps/intermediates. Could this be tested by labeling (e.g., in collaboration with Prof. Jeroen Dickschat)? Part of the concern is that the chosen arena for prediction, predictive analysis of STSs from Citrus bergamia, provided only modest (at best) results. In particular, of the three STSs predicted to proceed via N1 that were amenable to experimental characterization, apparently only two actually did so. Moreover, one of these two only had 11% predicted nerolidyl specificity! This seems extremely low, and begs the question of what the predicted specificity was for the STS that apparently did not proceed via N1 (this data is not presented but needs to be, both prediction and actual products). It would also be helpful to present the distribution of all STSs (and just those from C. bergamia) for predicted nerolidyl specificity, to better judge how well this approach does in distinguishing between those that carry out even this initial carbocationic transformation. In any case, the results are well short of having the sort of resolution that would be desired for any sort of accuracy in predicting STS catalytic activity (i.e, product outcome), which would require drilling down several additional steps in the catalyzed reaction. On the other hand, if more evidence can be provided supporting the claimed accuracy for predictive separation of STSs proceeding via N1, this would nevertheless represent a reasonable advance in this area.

**Have all data underlying the figures and results presented in the manuscript been provided?**

Reviewer #1: Yes

Reviewer #2: **No: **See comments regarding presentation of predicted specificity.

PLOS authors have the option to publish the peer review history of their article (what does this mean?). If published, this will include your full peer review and any attached files.

Reviewer #1: No

Reviewer #2: No
---

## [Decision Letter · Decision Letter 1]

15 Feb 2021

Dear Ms. Durairaj,

We are pleased to inform you that your manuscript 'Integrating structure-based machine learning and co-evolution to investigate specificity in plant sesquiterpene synthases' has been provisionally accepted for publication in PLOS Computational Biology.

Best regards,

Greg Tucker-Kellogg, PhD

Associate Editor

PLOS Computational Biology

Arne Elofsson

Deputy Editor

PLOS Computational Biology

Reviewer's Responses to Questions

**Comments to the Authors:**

Reviewer #1: I have reviewed the revised manuscript. The authors included additional data and made other minor revisions according to my suggestions. I have no further concerns.

Minor comment: The sentence starting with "Three residues from Y376 lies the residue" could be revised to "The residue positioned three residues upstream (or downstream?) of Y376"...

Reviewer #2: While shying away from doing the requested experimental work in the context of this manuscript, the authors have made a not unreasonable attempt to address my concerns, and I am satisfied that this study is now suitable for publication.

**Have all data underlying the figures and results presented in the manuscript been provided?**

Reviewer #1: Yes

Reviewer #2: Yes

PLOS authors have the option to publish the peer review history of their article (what does this mean?). If published, this will include your full peer review and any attached files.

Reviewer #1: No

Reviewer #2: **Yes: **Reuben J. Peters

---

## [Editor Report · Acceptance letter]

17 Mar 2021

PCOMPBIOL-D-20-01299R1 

Integrating structure-based machine learning and co-evolution to investigate specificity in plant sesquiterpene synthases

Dear Dr Durairaj,

I am pleased to inform you that your manuscript has been formally accepted for publication in PLOS Computational Biology. Your manuscript is now with our production department and you will be notified of the publication date in due course.

With kind regards,

Alice Ellingham
